# Conceptual Modeling of Extended Collision Warning System from the Perspective of Smart Product-Service System

**DOI:** 10.3390/s22124654

**Published:** 2022-06-20

**Authors:** Chunlong Wu, Hanyu Lv, Tianming Zhu, Yunhe Liu, Marcus Vinicius Pereira Pessôa

**Affiliations:** 1School of Mechanical Engineering, Hebei University of Technology, Tianjin 300401, China; lhyyyyy610@163.com (H.L.); 15940670320@sina.cn (T.Z.); liuyunhe372@163.com (Y.L.); 2National Engineering Research Center for Technological Innovation Method and Tool, Hebei University of Technology, Tianjin 300401, China; 3Tianjin Key Laboratory of Power Transmission and Safety Technology for New Energy Vehicles, Hebei University of Technology, Tianjin 300130, China; 4Faculty of Engineering Technology, University of Twente, Drienerlolaan 5, 7522 NB Enschede, The Netherlands; m.v.pereirapessoa@utwente.nl

**Keywords:** smart product-service system, extended collision-warning system, TRIZ function model, conceptual modeling

## Abstract

While Product-Service Systems (PSS) have a potential sustainability impact by increasing a product’s life and reducing resource consumption, the lack of ownership might lead to less responsible user behavior. Smart PSS can overcome this obstacle and guarantee correct and safe PSS use. In this context, intelligent connected vehicles (ICVs) with PSS can effectively reduce traffic accidents and ensure the safety of vehicles and pedestrians by guaranteeing optimal and safe vehicle operation. A core subsystem to support that is the collision-warning system (CWS). Existing CWSs are, however, limited to in-car warning; users have less access to the warning information, so the result of CWS for collision avoidance is insufficient. Therefore, CWS needs to be extended to include more elements and stakeholders in the collision scenario. This paper aims to provide a novel understanding of extended CWS (ECWS), outline the conceptual framework of ECWS, and contribute a conceptual modeling approach of ECWS from the smart PSS perspective at the functional level. It defines an integrated solution of intelligent products and warning services. The function is modeled based on the Theory of Inventive Problem Solving (TRIZ). Functions of an ECWS from the perspective of smart PSS can be comprehensively expressed to form an overall solution of integrated intelligent products, electronic services, and stakeholders. Based on the case illustration, the proposed method can effectively help function modeling and development of the ECWS at a conceptual level. This can effectively avoid delays due to traffic accidents and ensure the safety of vehicles and pedestrians.

## 1. Introduction

Under the background of a new round of information technology and scientific and technological revolution, all walks of life in China have entered a stage of high-quality development. While Product-Service Systems (PSS) have a potential sustainability impact by increasing a product’s life and by reducing resources consumption, the lack of ownership might lead to less responsible user behavior. Solving this problem requires changing the PSS value proposition [1], where a smart PSS can overcome this obstacle and guarantee correct and safe PSS use.

With the development and application of information and communications technology (ICT) such as the internet of things (IoT) [2,3,4] and digital twin (DT) [5,6,7], smart connected products (SCPs) have transformed servitization development in the manufacturing industry via the use of smart Product-Service Systems (smart PSS) [8,9,10].

As a promising application of smart PSS, collision-warning systems (CWS) of intelligent connected vehicles (ICVs) have the potential impact of reducing environmental pollution and traffic accidents and transforming economic development, thus contributing to sustainable development. As the number of vehicles has increased, driving safety has become a key problem to be solved urgently. Therefore, most vehicles are equipped with Advanced Driving Assistance Systems (ADAS) [11] to reduce traffic accidents. As the CWS is an important part of ADAS [12,13,14], conceptual modeling is particularly important from the perspective of smart PSS.

For the conceptual modeling, Zhou et al. modeled a functional digital twin for intelligent products based on the TRIZ function model [6]. Wu et al. proposed a conceptual model of the smart PSS based on the TRIZ function model and used the intelligent vehicle as an example to verify the method’s effectiveness [15]. Zhang et al. introduced the intelligent conceptual design for a complex electromechanical system using the TRIZ function model and function clipping and verified the method using a water surface platform [16]. Chang et al. developed a conceptual design process for a prosthetic hand using the TRIZ contradiction matrix and innovation principle and proposed the concept of a dexterous finger mechanism [17]. Kamarudin et al proposed a conceptual design method by combining constraints and TRIZ [18]. Liu et al. built an integrated interaction model of the product innovation based on TRIZ and bionics and used the AHP method to evaluate the solution [19]. Huang et al studied a process model of the product concept based on TRIZ and evaluated solutions using AHP to find an optimal product solution [20]. Liu et al. proposed the conceptual product innovation based on the TRIZ and functional analogy and Axiomatic Design in the development of an ultraviolet lampblack purifier and seedling clamping manipulator [21,22]. Therefore, as a tool for conceptual modeling, the TRIZ function model provides an effective method for the conceptual modeling of technical systems.

This paper aims to fill this gap by modelling the extended CWS as a smart PSS. A comprehensive process for developing a smart PSS conceptual model is presented, which results in an integrated CWS PSS solution including intelligent products and warning services. The process is based on the TRIZ function model technique. From the function realization, the conceptual scheme improves the existing CWS in its functional composition. As a result, stakeholders obtain an integrated solution combining products and warning services.

This research contributes:to PSS and smart PSS research by presenting a stepwise method to create a smart PSS conceptual design.to CWS research, (i) by presenting a novel understanding and the conceptual framework of the extended CWS from the perspective of smart PSS; (ii) a conceptual extended CWS modeling approach integrating an intelligent product system, stakeholders, and collision warning service system module; and (iii) a collision-warning service system to assist drivers and pedestrians in avoiding vehicle collisions and vehicle–pedestrian collisions.to the design for sustainability research by exemplifying how a smart PSS can help to guarantee the correct and safe use of PSS and thus increase its impact on sustainability.

The rest of this paper is organized as follows: Section 2 introduces the state-of-the-art smart PSS and extended CWS from the perspective of smart PSS. Section 3 expounds the construction process of the conceptual extended CWS from the perspective of smart PSS. Section 4 illustrates the concept with examples and a discussion was provided to analyze the whole study subsequently. This is followed by the conclusions of this study in Section 5.

## 2. Literature Review

### 2.1. Definition and Composition of Smart PSS

Smart PSS appeared in 2013. Valencia et al outlined six significant features of smart PSS: consumer empowerment, individualization of services, community feeling, service involvement, product ownership, and individual/shared experience, all of which could help designers create smart PSS development tools to effectively integrate products and services [23]. Kuhlenkötter et al. considered the smart PSS as an ecological service system in a digital form with highly complex, dynamic, and interconnected features [24]. Zheng et al. [8,9] thought SCPs to be communication portals connecting manufacturers, service providers, and users, impelling IT-driven and DT-enabled service innovation. The development process of the IT-driven PSS can be divided into three main stages: the internet-based PSS, IoT-enabled PSS and smart PSS. Key elements of the smart PSS can be extracted from numerous studies at three levels: the product-service level, the system level, and the system-of-systems level. Chowdhury et al. defined a smart PSS with digital dynamic capabilities in design and redesign including intelligence capability, connectivity capability, and analysis capability [25]. Chang et al. proposed a smart PSS development method from a user-centered perspective [10], as shown in Table 1.

From Table 1, the smart PSS endows the digital connection between the product and service systems with intelligent characteristics to realize user demands through interactions. Although there are different definitions, a smart PSS is an organic combination of intelligent products and e-services used to meet user demands for sustainable development.

Smart PSS is the overall solution of an intelligent product-service package. Smart PSS with complex functions is divided into modules [26,27] composed of stakeholders, intelligent product systems, and intelligent service systems to form the conceptual model of the extended CWS. The integration relationship among them is simply described as the Stakeholders–Intelligent Product–Intelligent Service system, as shown in Figure 1.

Stakeholders are an important part of the overall solution. Manufacturers perceive user demands, initiate new product development, and integrate suppliers to form the service content for user demands. The product architecture is designed with an appropriate interaction mode and interface. Further user demands will trigger improvements in product iterations and generate new services [10]. The product/service is a platform for users, manufacturers, and suppliers to obtain, transfer, and exchange values and realize co-creation.

The intelligent product system bridges stakeholders and the intelligent service system. Compared to the traditional product system, it gathers the intelligent perception, analysis and decision, control and execution, and intelligent interaction with its active perception of the environments and timely responses to change [28]. Applications of emerging information technologies such as big data, cloud computing, and IoT [3,4,29] present new features for autonomous learning, online upgrading, and human–machine integration.

The intelligent service system realizes the incremental value of the intelligent product system. Relevant services will also be endowed with new content due to artificial intelligence and big data [30]. Through the implementation of remote services and fault diagnosis systems, enterprises remotely obtain complete operation data from users for product design improvement, fault diagnosis, and user service. The competitiveness of enterprises can be effectively enhanced.

### 2.2. Smart Product–Service System Development

For development of the smart PSS, Poeppelbuss et al. [31] proposed an analysis and design tool based on an intelligent service canvas for the demand of lightweight tools. Wang et al. built a data-driven graphics demand extraction framework in the smart PSS context and extracted potential demands combined with the context-aware demands in a value co-creation manner [32]. Lee et al. proposed a structural service innovation method based on the enhanced TRIZ and service blueprint [30]. Liu et al. proposed the industrial smart PSS solution based on cyber–physical systems (CPS) and industrial IoT from the cyber–physical and edge-cloud perspectives [4]. Watanabe et al. developed an evolutionary design framework of smart PSS based on service engineering to overcome the limitation of available digital technology data in dynamic application environments [33]. Liu et al. proposed a comprehensive evaluation method based on the rough best–worst method (BWM) and TODIM (an acronym in Portuguese of Interactive and Multi-Criteria Decision Making) from the sustainability perspective of smart PSS and considered the bounded rationality and the judgment vagueness of decision makers [34]. Wu et al. (2020) built a conceptual model of smart PSS based on the TRIZ function model [15].

### 2.3. CWS Development

Recent research on the existing CWS focused on improving its hardware and algorithms, but it did not present a broader perspective including the system’s products and services. From the hardware-oriented aspect, Yu et al. proposed an embedded advanced CWS based on the monocular vision algorithm and high-speed processor to detect and send warning signals to the vehicle ahead for driving safety [35]. Chen et al. proposed an on-board navigation system based on data fusion, such as GPS, odometers, and Dedicated Short-Range Communication (DSRC), to obtain the adjacent vehicles’ information through network communication [36]. Gunasinghe et al. deployed CWS with three different types of sensors, simulated and compared their performance, and verified their reliability [37]. From the algorithm-oriented aspect, Dabbour and Easa proposed a perception framework and algorithm of CWS [38]. Milanés et al. proposed a rear-end collision-avoidance method based on the fuzzy controller of CWS [39]. Zhao et al. [40], Lee et al. [41], and Santos-González et al. [42] built different CWSs based on ICT and intelligent terminals. Yet, the functional dimension is less considered. Zhu et al. suggested that a forward CWS with the time-headway monitoring function could improve traffic efficiency using vehicle-to-vehicle communication technology for future transmission [43]. In addition, some research is based on analyzing and predicting driver and vehicle behaviors [44,45,46,47]. Lee et al. developed a rear-end collision-warning algorithm based on a multi-layer perceptron neural network, which is a real-time CWS that is not affected by human PRT through sensitivity analysis [48]. His algorithm is superior to previous algorithms in predicting potential dangers and can detect deceleration in advance. Huang et al. used Kalman filter technology to characterize the errors in position estimation and trajectory prediction and took the communication errors as part of the prediction errors, which improved the positioning reliability of CWS and the reliability of inter-vehicle communication [49]. Lee et al. proposed a CWS based on drivers’ driving behavior and used an artificial neural network learning algorithm to create a driving behavior model to determine the risk of collision, which improved the applicability of CWS [50]. Lee et al. proposed a back-end CWS framework based on an artificial neural network to consider THE PRT effect [51]. This algorithm is superior to other traditional algorithms in detecting and predicting the risk of rear-end collisions. This algorithm can be used for rear-end-collision warning and is not affected by human PRT. Kim et al. proposed an integrated module for lane and vehicle detection that can be embedded in the forward collision-warning system in the real-time operation of autonomous driving systems, improving the applicability of advanced driver assistance systems [52].

While most existing works focus on algorithms or evaluation methods, few studies work at the functional level. Most of the existing methods consider vehicles with ADAS, which usually require expensive sensors. However, pedestrians, bicycles, and other vulnerable road users are not easily accessible, and they have less access to the warning information. Studies have shown that most accidents are caused by drivers’ slow reactions at intersections, and the CWS should be gradually changed from camera/radar-based to communication-based systems. Therefore, it is necessary to provide helpful services to stakeholders such as drivers and pedestrians in order to present a broader range of holistic solutions including products and services.

In this paper, we apply previous research on the conceptual modeling of the extended CWS proposed by Wu et al. with the following reasons. On the one hand, Wu et al. modeled the conceptual design of the smart PSS at the functional level for a reference of the smart PSS development [17]. On the other hand, due to its related elements and characteristics, the extended CWS is a good candidate of the smart PSS for the formation of integrating intelligent products and services.

## 3. Conceptual Modeling of Extended CWS from the Perspective of Smart PSS Based on TRIZ Function Model

For the conceptual extended CWS from the perspective of smart PSS, this paper uses the TRIZ function model to search for solutions. This section briefly introduces the extended collision-warning system from the perspective of the smart product service system and the theoretical background and proposes the conceptual modeling process of extended CWS accordingly.

### 3.1. Extended CWS from the Perspective of Smart PSS

A CWS from the perspective of smart PSS, namely an extended collision-warning system, integrates sensing technologies, ICT, cloud computing, and other scientific technologies. A vehicle is equipped with a variety of sensors (e.g., vision sensors), controllers, and actuators for information exchange through the vehicle-to-everything (V2X) communication [53]. The extended CWS can be seen as an integrated solution consisting of intelligent products, stakeholders, and electronic services.

The extended CWS, from the perspective of smart PSS, consists of three layers, the sensing layer, network layer, and application layer, as shown in Figure 2. The sensing layer collects and transmits the information of stakeholders, vehicles, and roads to the network layer. The sensing layer is mainly composed of smart terminals, surveillance system, a road-sensing system, and a vehicle-sensing system. Smart terminals provide pedestrians with warning service information, the surveillance system provides real-time road conditions for relevant departments, the road-sensing system provides real-time traffic information for vehicles, and the vehicle-sensing system integrates on-board sensors and uploads data to the cloud in real time.

The network layer processes the information from the sensing layer, solves the collision warning algorithm in real time through cloud computing, and sends relevant service instructions to the application layer. At the network layer, pedestrian–vehicle–road–cloud communication is realized by network communication technology. The cloud server processes real-time location parameters of vehicles and pedestrians from the sensing layer, determines the collision possibility between vehicles and pedestrians, and provides the safe warning service [54,55,56]. In addition, there are important data of stakeholders, vehicles, and roadside equipment in the network layer for system security.

The application layer provides services based on information from the network layer. To avoid accidents, the application layer sends warning information to vehicles and pedestrians in real time once it receives the warning instructions from the network layer. If pedestrians and vehicles fail to respond in time and become congested, the transport agency will promptly send traffic information to the electronic information board. Other vehicles can make appropriate adjustments based on nearby conditions to maximize traffic efficiency through vehicle scheduling and stakeholder collaboration.

### 3.2. TRIZ Function Model

As an analysis tool, the TRIZ function model can be used to analyze the impact of system components and their interrelationships, characteristics, and effects in the system. A TRIZ function model consists of supersystems, components, products, and their interactions [55,56], as shown in Figure 3. The main steps of the function modeling are as follows.

(1) Determination of the system tree. The tree decomposition structure model, namely the function tree model, shows the total functions, subfunctions, and functional units of the system.

(2) Determination of the components, products, and supersystem. Components are parts of the system. The product is the goal to achieve, namely the system output, which is included in the supersystem.

(3) Interaction analysis. After the element is determined, the interaction analysis between components will be carried out to analyze the interaction types, namely the standard effect, the insufficient effect, the excessive effect, or the harmful effect, through the influence of one component on another.

(4) Graphical representation. Based on the interaction analysis, components are connected by arrows to form a functional model graphic of all components included in the system. Otherwise, Step (2) is performed again to find relevant components to completely present the required functions of the technical system.

### 3.3. Conceptual Modeling of Extended CWS

A conceptual modeling method is proposed for the extended CWS from the perspective of smart PSS based on the TRIZ function model. Specifically, the overall framework presents three modules: the intelligent product system, the stakeholders, and the warning service system, and their interactions. Then, the cloud platform elements and framework are introduced accordingly. The construction process is presented in Figure 4.

Functions are modeled for three modules of extended CWS: the intelligent product system, the stakeholders, and the warning service system module, with the following contents.

#### 3.3.1. Function Analyzing and Decomposing

Based on the intuitive features of the function tree, the total function of the technical system is decomposed. The subfunctions required are determined for the total functional realization and further decomposed to functional units, as shown in Figure 5.

#### 3.3.2. Function Module Division

The system is divided into function modules from the perspective of the smart PSS, namely the intelligent product system, the stakeholders, and the collision warning service system.

Defining the intelligent product system module: As carriers of warning services, corresponding intelligent products are considered for the pedestrian–vehicle–road–cloud information transmission, including intelligent vehicles, intelligent roadside equipment, the cloud server, and user terminal equipment.

Identifying the stakeholders involved in warning services: More focus is gradually placed on users of smart PSS, so more consideration should be given to user-centered CWS. Drivers and pedestrians have been considered more in the past, while in the era of IoT, the scope of stakeholders is considered to be broader and more comprehensive. For example, the traffic agency supervises roadside equipment, and the communication department provides communication services to stakeholders.

Building the warning service system module: This is realized through the interaction with stakeholders, considering the information transmission between vehicles and pedestrians. Warning instructions are generated through cloud computing for vehicles and pedestrians with collision tendencies in real time. In addition, communication security involving vehicle–road–cloud and visual services facilitated through terminals are considered.

#### 3.3.3. Modeling Partitioned Function Modules

The intelligent product system, stakeholders, and collision warning service system are based on the following processes.


**(1) Function modeling of the intelligent product system**


For an intelligent product system, the component composition and effect are decided at the functional level. The modeling process is as follows.

Step 1: Determination of the components and supersystem. Based on Step 1, the functional unit is used to determine the implementation carrier, namely the component. In addition, the environmental composition outside the technical system is considered for the extended CWS, namely the supersystem chosen to interact with the technical system.

Step 2: Interaction analysis. Based on the element composition, the interactions between components are decided for the interaction type, namely the standard effect, the insufficient effect, the excessive effect, and the harmful effect, by analyzing the component–component effect and the component–supersystem effect. Harmful or insufficient effects facilitate targeted improvements of the system.

Step 3: Graphical representation and improvement of the function model. Components are connected by arrows to form the function graphic model to fully present the required functions of the system. If not, Step1 should be performed again to determine the composition of relevant components.


**(2) Function modeling of stakeholders**


Stakeholders are core elements in the extended CWS for the user-centered smart PSS [11]. The function modeling process is as follows.

Step 1: Determination of the products and supersystem. Based on the stakeholders identified in Step 2, elements are modeled for interactions, namely products and supersystems.

Step 2: Interaction analysis. The interaction type of product–product or product–supersystem is analyzed for their influence on each other. It is necessary to reflect the category of relationship in the function model. In addition, the interaction type is determined according to whether the interaction relation can meet the demand between products.

Step 3: Graphical representation and improvement of the function model. The function model of the products and the supersystem is graphically expressed according to their interaction relationship, and the relationship between them needs to be briefly explained. A final verification is then performed on the missing elements to complete the model.


**(3) Function modeling of the collision warning service system**


A core element of the collision warning service system is the data flow to serve the stakeholders with the warning service information. The function modeling process is as follows.

Step 1: Functional analysis. The warning service system receives data instructions from the intelligent product system and sends the relevant instructions to terminals, such as stakeholders, after processing. Specifically, it combines the intelligent product system and the stakeholder module, takes the user as the starting point, and defines the required functions. In addition, the analysis of the intelligent product system modules with big data can promote the further development of services and generate new value opportunities. With the development and application of ICT, the CWS gradually changes from sensor-based warnings to communication-based warnings. Therefore, the extended CWS is considered. For example, a blocked vehicle can predict the forward information based on communication.

Step 2: Determination of components and products. When constructing the function model of the collision-warning service, the components and products will be mainly considered. Through functional analysis combined with user service requirements, the realization mode and carrier of the service are determined for the final components and products.

Step 3: Interaction analysis. Through the action form analysis of the component–component and component–product relationships, the interaction types between them are determined.

Step 4: Function model improvement. The function models of the products and supersystems are illustrated according to their functional relations, and the interactions between them are briefly explained. The function models are evaluated and improved to fully present the required functions.

#### 3.3.4. Forming Network Platform of the System 

The application of the SCPs and sensors will generate large amounts of user data, product data, and service-related data. Through access to the device, data can be collected by the terminal interface and sent to the cloud platform. The database collects and stores multi-source data. Specifically, it includes basic and location information of users, basic information of the devices (e.g., vehicle VIN code, devices number), the running states (e.g., velocity, status), the location information of devices, and the service data, which involves data management, warning service, and event storage.

For an effective warning system, the length of the data transmission link is reduced to improve the timeliness and edge cloud service [30]. The near-field processing uses the road-end equipment close to the vehicle. The collision-warning algorithm works in real time through edge computing and other technologies, and the results are fed back to drivers and pedestrians for a timely warning.

Data analysis and data storage are then performed. Specifically, data analysis cleans and visualizes the data flow between the components. The important data storage mainly includes the vehicle’s track history and the parameter history data playback (e.g., velocity, acceleration) for the traceability of important events. Moreover, the mass of data in the cloud platform by service interface can be mined and analyzed to promote the generation of new services. For example, a section of the road often produces collision-warning information, and the transport agency intentionally guides users to avoid accident-prone areas. As shown in Figure 6.

#### 3.3.5. Marking the Overlap of Function Modules

The extended CWS from the perspective of smart PSS is composed of the intelligent product system, stakeholders, and the warning-service system module. However, these three modules are not independent, but rather an organic whole that is completely connected. When modeling them separately, overlapping parts are marked for subsequent overall modeling.

#### 3.3.6. Building the Overall Function Model of Extended CWS

Labeled models are merged to form the conceptual modeling of the extended CWS from the perspective of the smart PSS as shown in Figure 7. Details are as follows.

The device-sensing layer reports the vehicle or environment status data through the on-board or roadside equipment and transfers location data and other information through the user terminal.

The network transport layer ensures the real-time performance of data transmission. High-speed data transmission is realized through the transitional network connection. The completion of a safe and reliable transmission of the control link guarantees the low delay of data transmission.

The data-processing layer includes near-field processing and cloud processing. Near-field processing is for road-end devices near vehicles to reduce the length of the data transmission link and improve the timeliness for low latency requirements. The data-processing layer contains multiple types of database, including environmental awareness, decision algorithms, and device state databases.

The application layer includes the security service, the communication service, the supervision service, the visual service, and the warning service. Users’ multiple service requirements are achieved through the human–machine interaction design for “pedestrian–vehicle–road–cloud” efficient coordination.

The user layer is the interface for user interactions with the cloud platform, mainly for management departments, service providers, drivers, and pedestrians.

## 4. Case Study

The extended CWS from the perspective of the smart PSS integrates human, vehicle, environment, and service to improve driving safety and traffic efficiency and prevent traffic accidents. This case study develops the extended CWS with the intelligent vehicle Dongfeng S50 and sightseeing vehicle in the intelligent driving experimental area (IDEA) of the China Automotive Technology and Research Center (CATARC), as shown in Figure 8 and Figure 9.

### 4.1. Conceptual Model Development

#### 4.1.1. Extended CWS from the Perspective of Smart PSS

In terms of composition, the stakeholders mainly include the relevant management departments, the drivers, and the pedestrians, as shown in Figure 10. The intelligent product system includes ICVs, roadside equipment, cloud platform, data acquisition equipment, etc. The collision-warning service system provides services of security, communication, supervision, visualization, and warning. In particular, the supervision service monitors the running status of vehicles and roadside equipment for the release of important information and vehicle scheduling, which play a crucial role in the entire system.

Considering the interfaces:the intelligent product system provides a convenient travel platform for stakeholders to receive warning services;the collision-warning service system provides high-quality travel services for stakeholders; andthe stakeholders utilize the warning service as a convenient travel platform for a high-quality travel service and provide feedback to the intelligent product system and collision-warning service system.

In terms of impact, the extended CWS from the perspective of smart PSS reduces users’ travel waiting time traffic accidents, improves travel efficiency, and ensures the safety of vehicles and pedestrians.

#### 4.1.2. Function Modeling of Extended CWS

**Step 1:** Function analysis and decomposition. The extended CWS from the perspective of the smart PSS provides an integrated solution of multiple elements, including tangible products, intangible services, and stakeholders. Based on the sensing-layer information from the vehicle, pedestrian, and roadside equipment, the network layer processes the data and collision possibility in real time via the cloud server, to send real-time warning instructions to vehicles and pedestrians.

All subfunctions cooperate in the warning service. The pedestrian–vehicle–road–cloud communication system works in real time on the cloud server based on the collected location information to provide real-time warning instruction services for vehicles and pedestrians. The system also traces important data. Therefore, the total function includes sub-functions such as V2X communication, information collection, collision calculation, warning service, and storage, as shown in Figure 11. V2X communication mainly includes Vehicle-to-Infrastructure (V2I) communication, Vehicle-to-Pedestrian (V2P) communication, and Vehicle-to-Vehicle (V2V) communication.

**Step 2:** Function module division. From the perspective of the smart PSS, the extended CWS has three modules: the intelligent product system, the stakeholders, and the collision-warning service system. In addition to ICVs, sensors, user terminal devices, roadside equipment (e.g., intelligent traffic lights, roadside units), cloud servers for cloud computing, base stations for positioning, and communication are included in the intelligent product system. For drivers and pedestrians, the transport agency manages roadside facilities, the communication department connects the vehicles and the road, the IT department manages and maintains the cloud platform, and the service providers are included in the stakeholder module. For the warning service system, the interactions of the drivers and pedestrians include voice prompts, data visualization, etc. The safety of the network and communication is ensured for the supervision service provided to the transport agency.

**Step 3:** Function module modeling. The TRIZ function models are built for the intelligent product system, stakeholders, and the collision-warning service system modules, respectively.


**(a) Function modeling of the intelligent product system**


The engineering system extended CWS is partitioned for the components and supersystems. Products in the supersystem components also exist outside of the engineering system, such as in the road environment and electric energy, as shown in Table 2. Components Analysis of Extended CWS is shown in Table 3.

Components, products, and supersystems, as well as their interactions, are connected by the corresponding arrows to form a function model of the intelligent product system as shown in Figure 12.


**(b) Function modeling of stakeholders**


Stakeholders include drivers, pedestrians, transport agencies, communication departments, IT departments, and service providers. In addition, the supersystem road environment provides the travel conditions for stakeholders, and the cloud platform collects and processes the data (e.g., personal info, location info) in the sensing layer for real-time warning instructions to the stakeholders. The function model of the stakeholders is shown in Figure 13.


**(c) Function modeling of the collision-warning service system**


Collisions usually occur between vehicles or between vehicles and pedestrians. We built the function model of the warning service system mainly based on these two aspects. The information sent from the sensing layer to the network layer is processed in Step 3 (a), and the modeling of the warning information function sent from the network layer to the application layer is followed.

First, Collision warning between vehicles. The cloud platform sends the warning instruction to the vehicle in danger of collision. The warning instruction is issued by the cloud platform and sent to the RSU via the LTE-V base station. The RSU sends the warning instruction to the OBU of the nearby vehicle S50 and to the OBU of the pre-collision vehicle for the driver to brake. This is performed through the vehicle’s audio/video interaction as shown in Figure 14.

Then, Collision warning between vehicles and pedestrians. As shown in Figure 15, a warning service instruction is sent to pedestrian 1; the vehicle can automatically sound its horn. In addition, if pedestrian 1 fails to respond to the signal, the cloud platform warns pedestrian 1 by making the mobile terminal of the nearest pedestrian 2 also emit a sound.

**Step 4:** Cloud platform supports data transmission, storage, management, and analysis.

As shown in Figure 16, the central cloud is connected to unified service platforms, including security, map, insurance, and other service platforms, to provide travel services. For example, high-precision maps provide real-time location services for users.

The regional cloud provides information along with regional level auxiliary services for transportation and communications. The regional cloud analyzes vehicle and pedestrian data in the database, intersection data with collision behaviors for management, and the scheduling of regional service departments.

In addition to the IoT service environments of a network, the edge cloud servers also provide real-time computing to reduce the computing power, power requirements, and latency of IoT devices [29]. In mobile edge calculation (MEC), users and IoT devices upload delay-sensitive computing tasks to edge servers, which process received computing tasks locally and feed the results back to mobile users and IoT devices. In addition, users and IoT devices wirelessly send real-time data from their devices to servers for storage, allowing the backend server to monitor the status of the devices in real time.

**Step 5:** Marking crossing parts. Steps 3 (a) and (b) include the supersystem road environments and system component cloud platform. Steps 3 (a) and (c) include the supersystem electric energy, the system components cloud platform, LTE-V base station, RSU, OBU, vehicle S50, and the user terminals. Steps 3 (b) and (c) include the system component cloud platform and stakeholders. These elements are marked for subsequent merging intersections.

**Step 6:** Function modeling of extended CWS. An overall framework of the system function model is shown in Figure 17.

### 4.2. Composition of Extended CWS and Discussion

#### 4.2.1. Composition of Extended CWS

The intelligent product system is based on the two ICVs as the platform. The test bench is shown in Figure 18. The sightseeing vehicle is the target vehicle, the Dongfeng vehicle is the test vehicle, and OBU is carried out to communicate with vehicles and roadside equipment based on V2X. The edge-cloud-computing results are fed back to the control center, and pedestrians are given collision avoidance service instructions through the mobile terminal.

In addition to drivers and pedestrians, other stakeholders in the collision scenario including service providers, IT departments, transport agencies, and communication departments are also considered. Service providers and IT departments manage and maintain the cloud platform, transport agencies manage the roadside equipment, and communication departments build and maintain communication facilities.

For stakeholders other than drivers and pedestrians, management and maintenance services are carried out based on the monitoring management platform, as shown in Figure 19. The real-time statuses of vehicles and roadside equipment are viewed in the interface. For drivers, visual services are provided based on Figure 20. The latitude, longitude, speed, and time to collision of the test and target vehicles are viewed in real time. For pedestrians, crossing the road is based on the road-guided service provided by the mobile terminal.

Location information

Parameters such as longitude, latitude and heading angle, of the Dongfeng vehicle and the sightseeing vehicle are interpreted based on CANape17.0 (Version 17.0, Vector, Germany). To provide visual services for drivers, the location information of the two vehicles is displayed in real time in *the Map Window* as shown in Figure 20. The sightseeing vehicle is represented by the red cross, and the Dongfeng vehicle is represented by blue triangle. The parameters of the vehicles are displayed in real time in the *Graphic window* and *Numeric window*. The longitude, latitude, and altitude of the sightseeing vehicle are represented by *GPS_x*, *GPS_y*, and *GPS_z*, respectively, and the longitude, latitude, and altitude of the Dongfeng vehicle are represented by *INS_Longitude*, *INS_Latitude*, and *INS_LocalHeight*, respectively. Visual services help drivers prepare to slow down in advance in a collision scenario.

**Figure 20 sensors-22-04654-f020:**
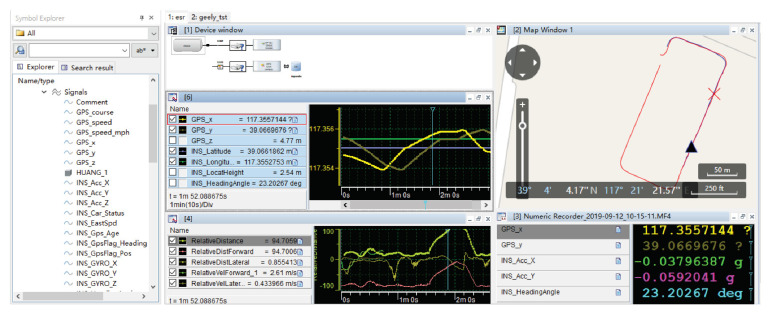
Visualization interface based on CANape17.0.

b.Distance information

The relative distance (*Relative_Distance*) of the two vehicles is calculated in real time using the programming language. The application shows the decomposition of relative distance (*Relative_Dist_Forward* and *Relative_Dist_Lateral*) as shown in Figure 21. In addition, the *Relative_Dist_Forward* and *Relative_Dist_Lateral* of the two vehicles are calculated with the help of the vehicle coordinate system, the relative distance, and the heading angle. The positive and negative properties of the curve represent the relative or opposite driving of the two vehicles. When vehicles or pedestrians intersect, drivers slow down ahead of time with the help of visual curves or digital information. In addition, based on the voice prompts and speed-guided services of the on-board iPad, the driver has sufficient time to brake before entering the collision scene. The extended CWS provide robust solutions to customers with bundles of “intelligent products” and “electronic services”.

c.Time-to-Collision information

Time-to-Collision (TTC), that is pre-collision time, is a key element in collision scenarios. It includes longitudinal TTC (*TTC_Forward*) and lateral TTC (*TTC_Lateral*). Taking *TTC_Forward* as an example, the *Relative_Dist_Forward* and driving time of the two vehicles are used to solve the *TTC_Forward* as shown in Figure 22. The service-oriented, extended CWS provides voice services and visual services for drivers and pedestrians. Other parameters can be added (e.g., the lateral/longitudinal velocity, acceleration) while taking into account the requirements of various stakeholders at the same time.

During the process of driving, the Dongfeng vehicle recognizes the traffic lights at intersections through in-vehicle cameras and directly transmits the statuses of traffic lights to the vehicle’s control system based on V2X, as shown in Figure 23.

Road information released by the electronic information board is transmitted to the Dongfeng vehicle based on V2X. In addition, the Dongfeng vehicle plans and adjusts the path in advance to reduce the waiting time on the road, as shown in Figure 24.

In the process of driving, the cameras and lidar detect pedestrians and transmit the identification result to the Dongfeng vehicle’s controller based on V2X, so as to realize pedestrian identification and avoidance in blind areas at intersections, as shown in Figure 25.

In the process of driving, the V2X equipment installed on the Dongfeng vehicle constantly obtains the position, speed, and direction information of the sightseeing vehicle, so as to realize the function of the collision warning at intersections, as shown in Figure 26.

In the process of driving, the Dongfeng vehicle realizes the detection and identification of the sightseeing vehicle in front of it through the millimeter-wave radar and lidar, thus realizing the forward collision-warning function, as shown in Figure 27.

#### 4.2.2. Discussion

Facing the development of smart technology in industry 4.0, smart PSS is a new and valuable extension of traditional PSS in servitization transformation. From the above description with the illustrative case study, it appears that the TRIZ function model theory can be utilized in the conceptual modeling of extended CWS from the perspective of smart PSS. The conceptual modeling of extended CWS from the perspective of smart PSS is meaningful to both academia and practice, as it provides support to the design and operation of smart PSS.

Theoretically, this study presents a novel understanding and a conceptual framework of extended CWS. In addition, this study has determined three key conceptual factors (i.e., intelligent product system, warning service system, and stakeholders) for extended CWS from the perspective of the smart PSS. Lots of previous similar research conducted algorithm optimization designs and evaluations of traditional CWS, but few had taken holistic account of the intelligent product system, warning service system, and stakeholders of CWS by starting from the perspective of the smart PSS. Compared with traditional CWS, this proposal considers more elements of a collision scenario from a broader perspective. This study considered all stakeholders in the scenario, which is an important factor that was not taken into account in previous relevant studies. After the cloud platform senses the information and processes it, it will send early warning information to stakeholders such as pedestrians or vehicles that have a pre-collision system, which can reduce the probability of a collision to a greater extent and make travel safer and more effective. The conceptual-factor-warning service system based on voice and visual services can fill the wide chasm that results in pedestrians receiving less warning information. It is helpful for enterprise managers to accelerate the transformation of future vehicle warning development modes as well as increase firm core competitiveness.

Practically, traditional CWS will be gradually replaced by extended CWS, when smart devices become more and more low-cost. As such, manufacturers will have to face fierce competition in the near future, as well as transform and develop towards an extended collision-warning system (extended CWS), which is taken as an environment hub to pursue healthy and happy travel. The derived practical implications of this study are as follows. First, based on IoT and ICT, a conceptual extended CWS, concerning pedestrians and other stakeholders, was fabricated using the TRIZ function model to help prescribe abstract solutions. Second, three types of conceptual factors (intelligent product system, warning service system, and stakeholders) on extended CWS were captured to form a multi-layer service structure. Third, by applying this proposed conceptual modeling approach to extended CWS, drivers and pedestrians received effective warning instructions based on voice and visual services. For example, based on the information in the *Graphic_window* and *Numeric_window* (Figure 20), the driver braked in advance to reduce the possibility of collision. In addition, the driver can also take advantage of the iPad’s speed guide and voice services (Figure 26 and Figure 27). Pedestrians can decelerate in advance based on the iPad with voice service (Figure 25). In short, this so-called conceptual modeling framework can help prioritize the CWS for smart PSS to support its development and operation management for practitioners.

## 5. Conclusions

This work aims to develop a conceptual modeling development approach of extended CWS from the perspective of smart PSS. For this purpose, a new understanding of extended CWS from the smart PSS perspective is provided, and a detailed development approach based on the TRIZ function model with an intelligent product service system modeling, warning service system modeling, and stakeholder modeling is presented. The extended CWS from the perspective of the smart PSS presents a comprehensive and extensive perspective. Specifically, vehicles, roadside equipment, and user terminal equipment are considered in the intelligent product system. Stakeholders other than drivers and pedestrians, service providers, the communication department, and the transport agency are considered. The realization of a warning service system depends on the joint efforts of all stakeholders. Voice and visual services based on the iPads used by pedestrians can make up for having less access to the warning information. Pedestrian–vehicle–road–cloud collaboration is realized based on V2X to form an extended CWS from the perspective of the smart PSS. Service-oriented extended CWS reduces the probability of accidents by planning routes in advance, ensuring the safety of vehicles and pedestrians, and reducing travel wait times.

In summary, this article can make following contributions to the research field of extended CWS for smart PSS. First, a novel understanding and the conceptual framework of extended CWS are presented. Second, a conceptual modeling approach for extended CWS from the perspective of the smart PSS is proposed. Third, compared to traditional CWS studies, the proposed framework can holistically cover conceptual factors such as the intelligent product system, the warning service system, and stakeholders for extended CWS from the perspective of smart PSS. Fourth, drivers and pedestrians can obtain effective voice and visual services to reduce the occurrence of collisions and achieve sustainability. Moreover, an illustrative case study of extended CWS is demonstrated along with this proposed methodological framework.

However, there are some limitations waiting for research in the future. Extended CWS from the perspective of the smart PSS in this article focuses on the integration of SCPs, e-services, and stakeholders and touches on fewer viewpoints that extend to platform services. Nevertheless, this work, as an explorative study, is still confined to theoretical research by only looking at a systematic process to realize the proposed modeling approach, and details such as the implementation costs have not been considered. Some comparative evaluations (e.g., judgment based on hesitant fuzzy/rough numbers) have not yet been carried out. Additionally, more empirical tests should be implemented to obtain validity and improve the practical applicability.

## Figures and Tables

**Figure 1 sensors-22-04654-f001:**
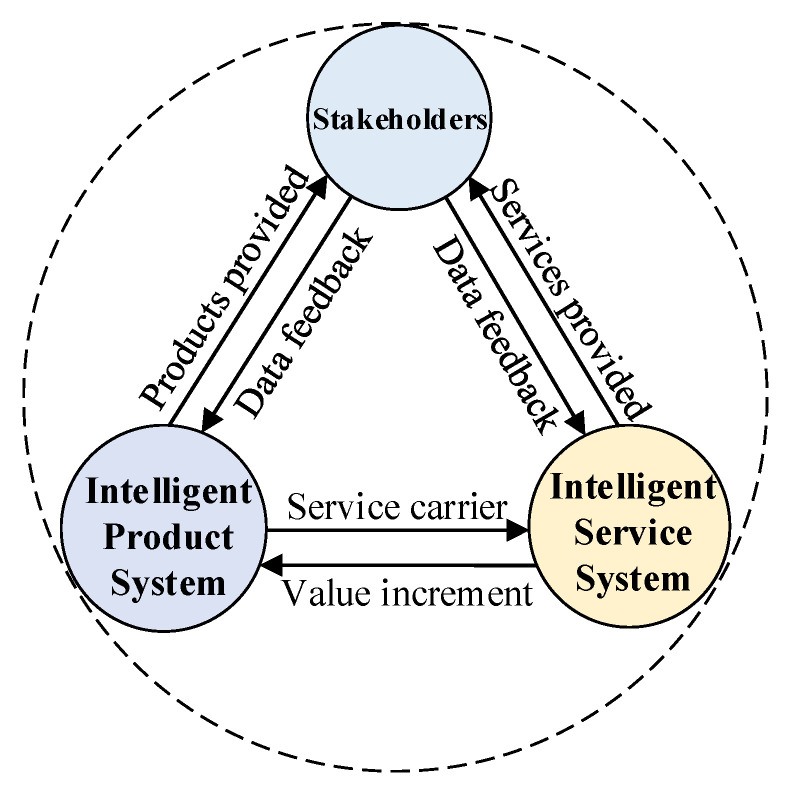
Stakeholders–Intelligent Product–Intelligent Service system.

**Figure 2 sensors-22-04654-f002:**
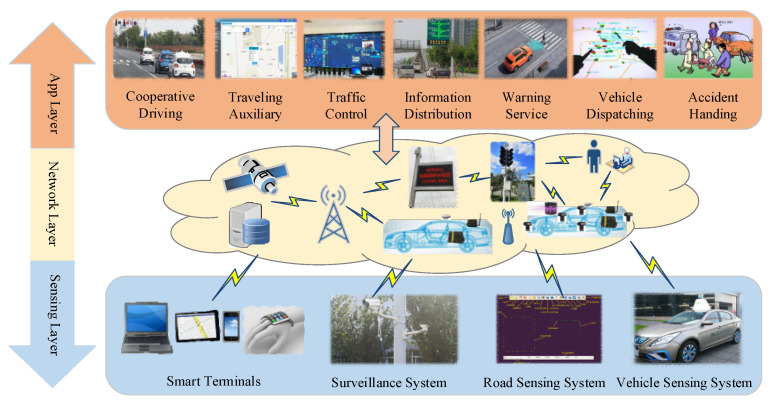
Conceptual framework of extended CWS.

**Figure 3 sensors-22-04654-f003:**
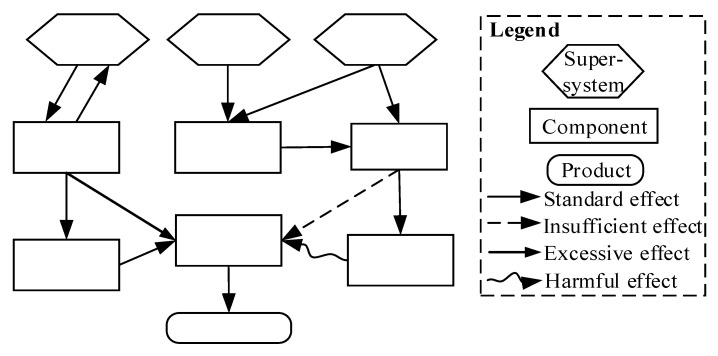
Elements of the TRIZ function model—fill contents in the chart.

**Figure 4 sensors-22-04654-f004:**
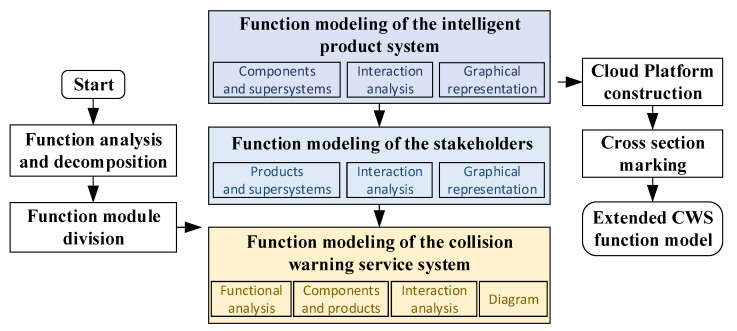
Proposed modeling process of the extended CWS.

**Figure 5 sensors-22-04654-f005:**
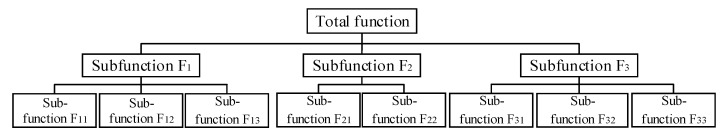
Functional decomposition process.

**Figure 6 sensors-22-04654-f006:**
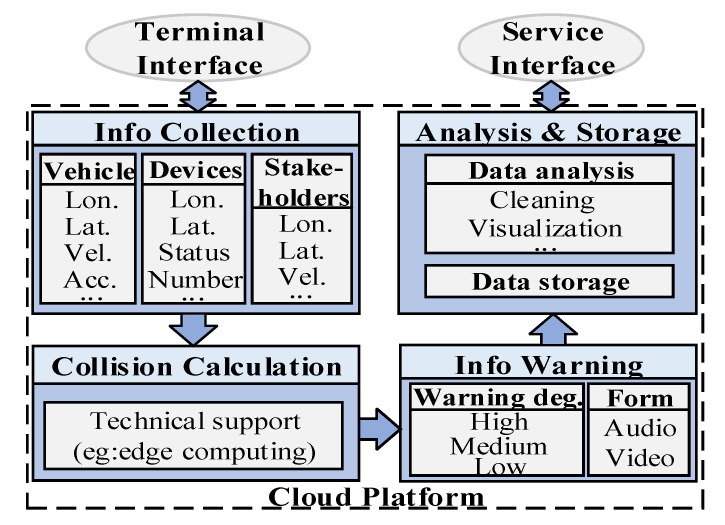
Cloud platform structure.

**Figure 7 sensors-22-04654-f007:**
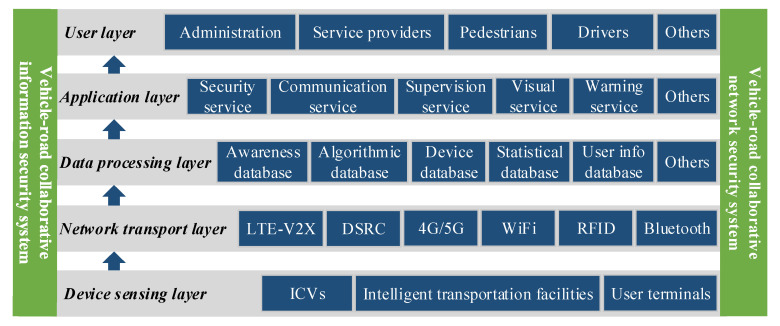
Overall architecture of extended CWS from the perspective of smart PSS.

**Figure 8 sensors-22-04654-f008:**
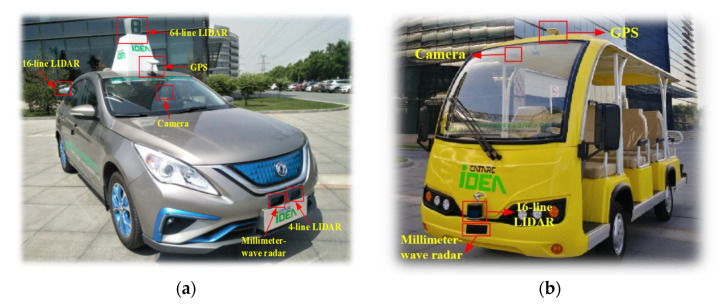
(**a**) Dongfeng S50 EV. (**b**) Sightseeing vehicle.

**Figure 9 sensors-22-04654-f009:**
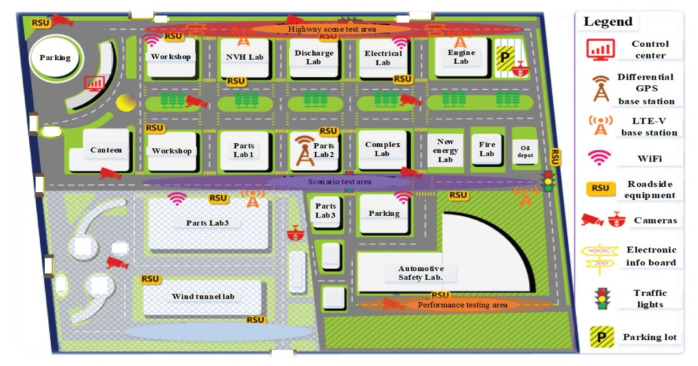
Experimental Area.

**Figure 10 sensors-22-04654-f010:**
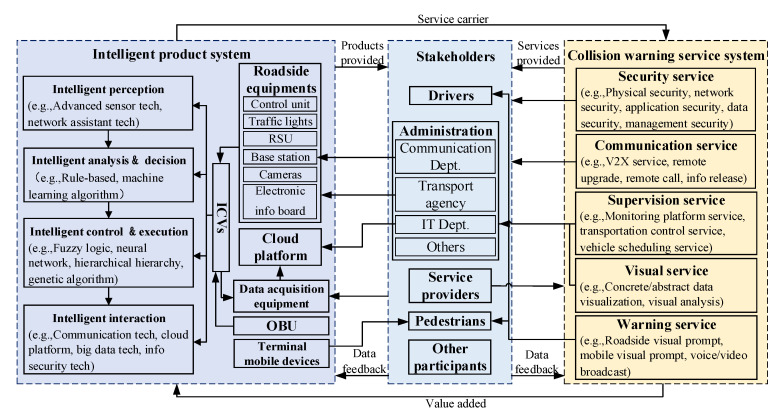
Extended CWS from the perspective of smart PSS.

**Figure 11 sensors-22-04654-f011:**
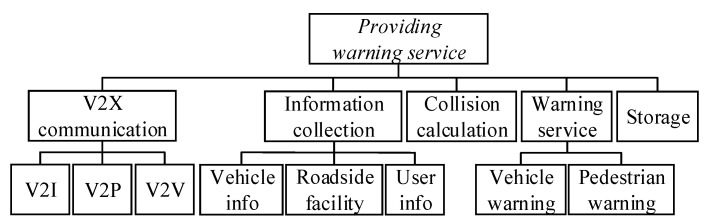
Functions of extended CWS from the perspective of smart PSS.

**Figure 12 sensors-22-04654-f012:**
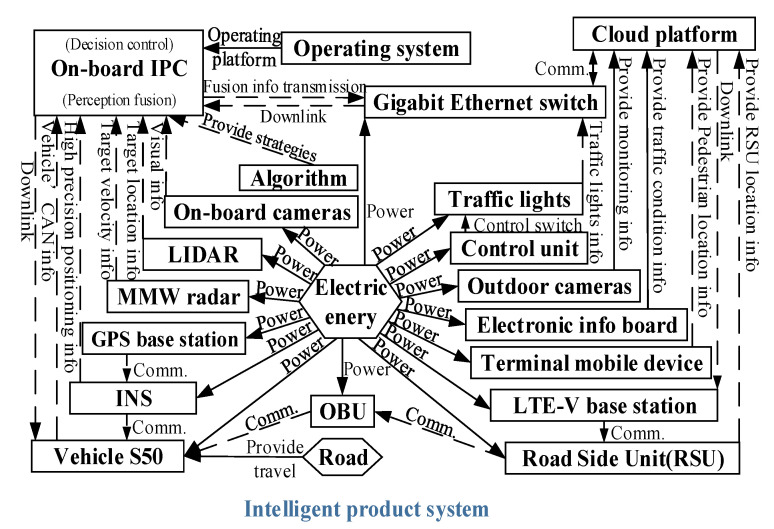
Function model of the intelligent product system.

**Figure 13 sensors-22-04654-f013:**
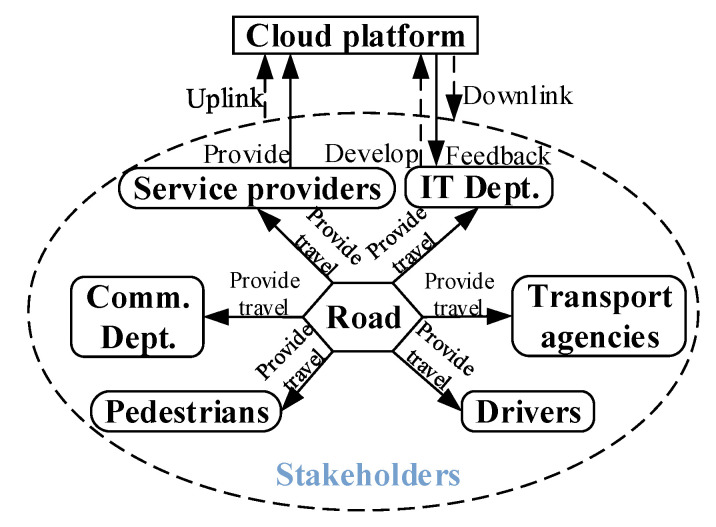
Function model of stakeholders.

**Figure 14 sensors-22-04654-f014:**
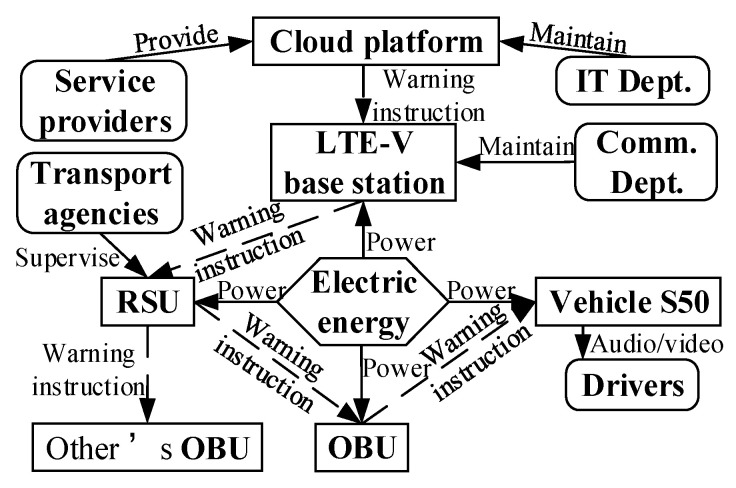
Function model of vehicle collision-warning service system.

**Figure 15 sensors-22-04654-f015:**
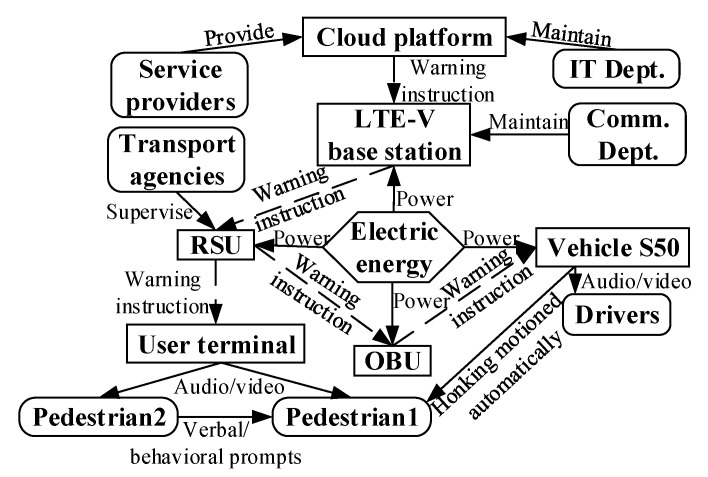
Function model of vehicle–pedestrian collision-warning service system.

**Figure 16 sensors-22-04654-f016:**
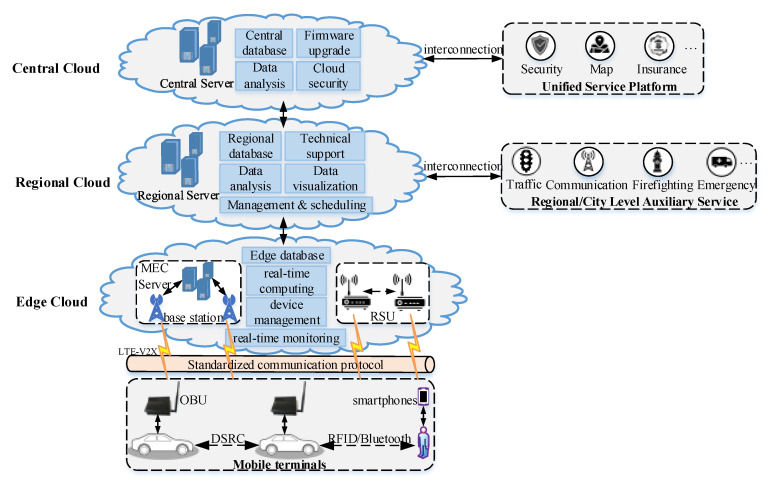
The cloud platform of the extended CWS.

**Figure 17 sensors-22-04654-f017:**
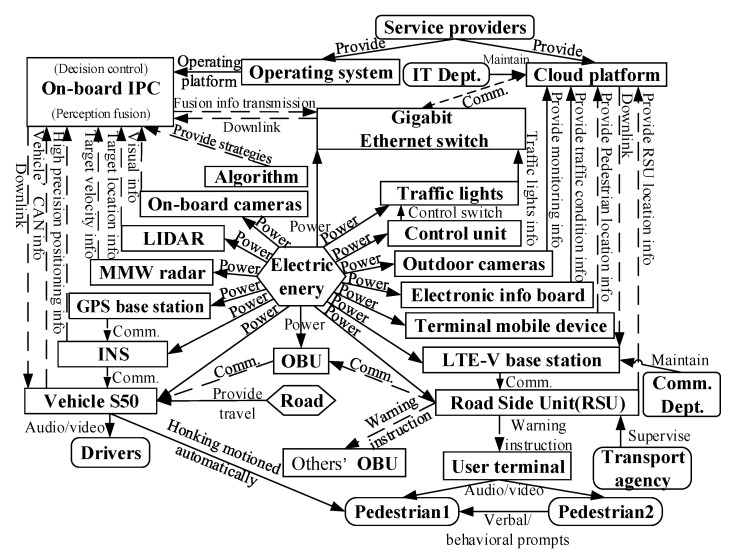
The function model of the extended CWS.

**Figure 18 sensors-22-04654-f018:**
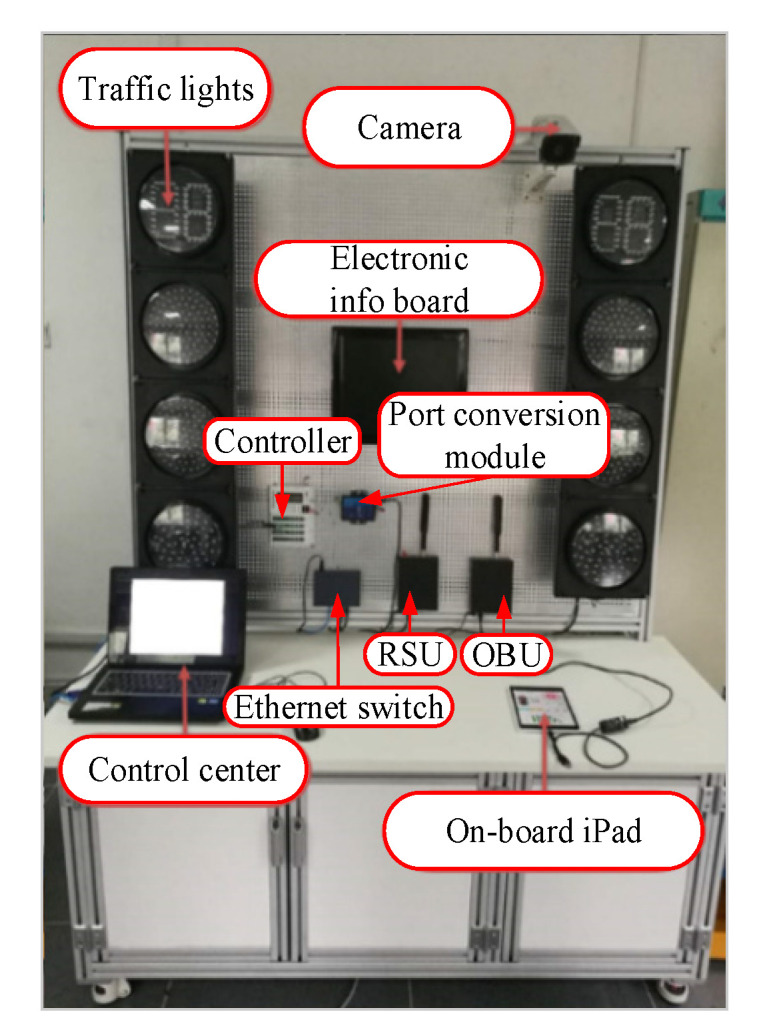
Parts of the indoor bench installation of the system.

**Figure 19 sensors-22-04654-f019:**
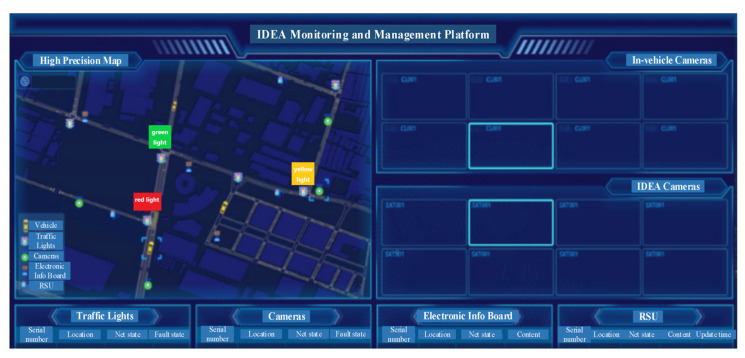
Monitoring platform.

**Figure 21 sensors-22-04654-f021:**
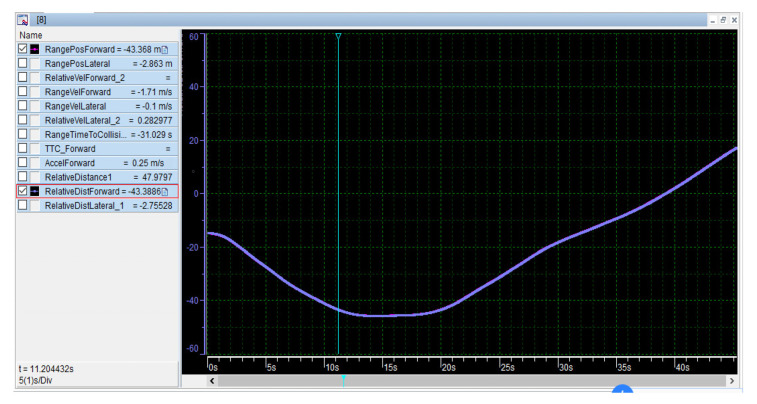
*Relative_Dist_Forward* of two vehicles.

**Figure 22 sensors-22-04654-f022:**
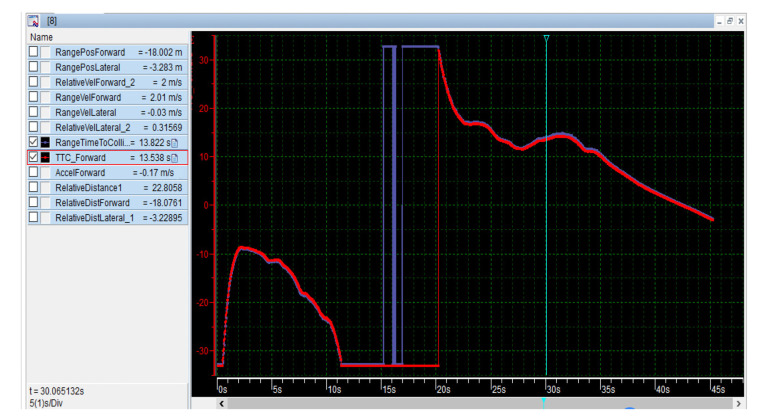
*TTC_Forward* of two vehicles.

**Figure 23 sensors-22-04654-f023:**
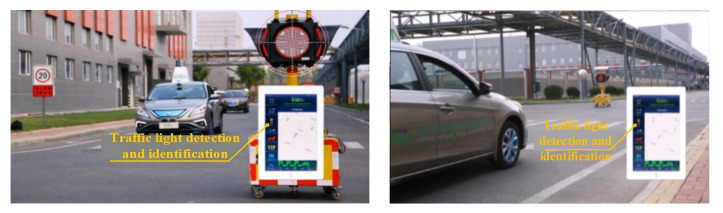
Traffic lights recognition based on V2X.

**Figure 24 sensors-22-04654-f024:**
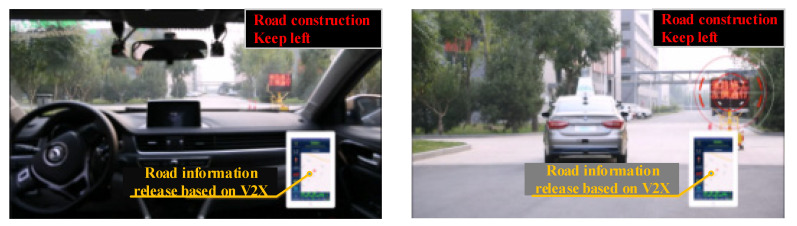
Road information publishing based on V2X.

**Figure 25 sensors-22-04654-f025:**
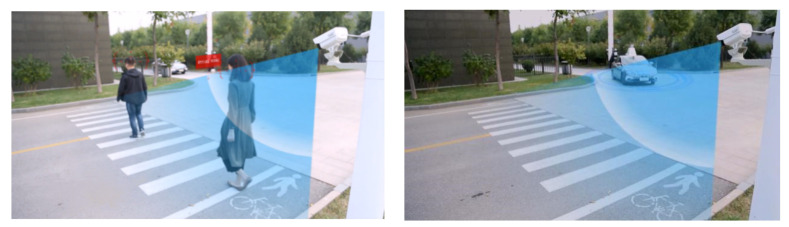
Pedestrian recognition and avoidance based on V2X.

**Figure 26 sensors-22-04654-f026:**
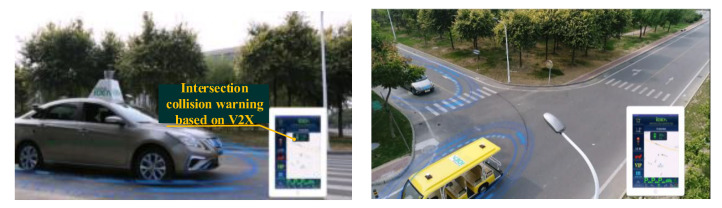
Intersection collision warning based on V2X.

**Figure 27 sensors-22-04654-f027:**
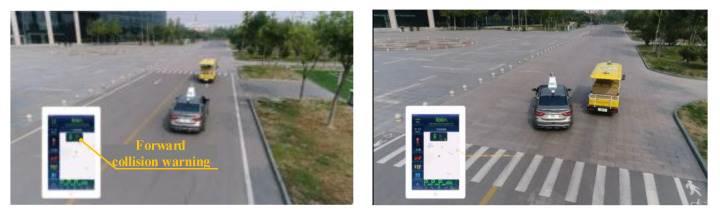
Forward collision warning based on V2X.

**Table 1 sensors-22-04654-t001:** Smart PSS definitions.

Year	The Author(s)	Definitions
2013	Valencia et al. [23]	Intelligent products and e-services are integrated into a single solution through ICT technologies.
2017	Kuhlenkötter et al. [24]	Digital connectivity between components enables autonomous interaction and further development of products and services as a result of digitalization.
2018	Zheng et al. [8,9]	A business strategy of IT-led value co-creation, involving different stakeholders as participants, smart systems as infrastructure, SCPs as the medium, and the e-services they generate as the primary value, is continuously committed to meeting customer needs in a sustainable manner.
2018	Chowdhury et al. [25]	A combination and interaction based on smart technologies, physical products, services, and business models to meet user demands.
2019	Chang et al. [10]	Product and service integration is achieved by integrating stakeholders and capturing user requirements in the physical environment and building databases in the cloud environment.

**Table 2 sensors-22-04654-t002:** Elements of extended CWS.

Engineering System	Components	Supersystems
The extended CWS	vehicle S50, Inertial Navigation System (INS), differential GPS base station, millimeter-wave radar (MMW radar), LIDAR, on-board cameras, on-board Industrial Personal Computer (IPC), operating system, algorithms, gigabit Ethernet switch, traffic lights, control unit, outdoor cameras, electronic information board, terminal mobile device, LTE-V base station, Road Side Unit (RSU), On-Board Unit (OBU), V2X communication system, cloud platforms, etc.	road environment, electric energy, stakeholders (drivers, pedestrians, transport agency, communication department, IT department, service providers, etc.)

**Table 3 sensors-22-04654-t003:** Components analysis of extended CWS.

Elements	Function Description	Interaction Type	Problematic Function
**Supersystem(s)**	road environment	Provides travel conditions.	standard	
electric energy	Power supply.	standard	
stakeholders	Provides relevant services and receives warning services.	standard	
**Components**	vehicle S50	Provides travel tools.	standard	
INS	Provides precision localization information.	insufficient	√
differential GPS base station	Precise localization.	insufficient	√
MMW radar	Provides the speed information of the target.	insufficient	√
LIDAR	Provides location information of the target.	insufficient	√
on-board cameras	Provides surrounding information to the cloud.	insufficient	√
on-board IPC	Realizes data fusion and decision control.	insufficient	√
operating system	Provides operating platform for IPC.	standard	
algorithms	Provides strategies for IPC.	insufficient	√
gigabit Ethernet switch	Realizes the information exchange between IPC, roadside facility, and cloud.	insufficient	√
traffic lights	Provides traffic signal information.	standard	
control unit	Implement control operation.	standard	
outdoor cameras	Provides traffic information.	standard	
electronic information board	Displays real-time traffic information.	standard	
terminal mobile device	Human-machine interaction.	standard	
LTE-V base station	Connects the cloud and the terminals.	insufficient	√
RSU	Real-time communication with OBU.	standard	
OBU	Real-time communication with RSU.	standard	
V2X communication system	Provides real-time communication.	insufficient	√
cloud platforms	Provides real-time cloud computing.	standard	
**Product(s)**	stakeholders	Provides relevant services and receive warning services.	standard

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
