# Peer review of "Conceptual Modeling of Extended Collision Warning System from the Perspective of Smart Product-Service System"

_sensors, 2022, doi:10.3390/s22124654_

Round 1
Reviewer 1 Report
I suggest a comparison of the results reported by the paper with the results of previous similar works.
Reviewer 2 Report
This paper presents a conceptual modeling approach for extended CWS from the perspective of smart PSS. The paper was written in a solid manner. The purpose of the paper its scope and scientific contributions are presented. A literature review has been conducted. The concept has also been supported with selected case studies. The authors have done a piece of solid work. For the future, the work can be improved by organizing it better. Some parts of the literature review have been included in the practical section (see the beginning of section 3.1.). This should have been included in the chapter related to the literature review.
Reviewer 3 Report
The authors' paper's main contribution is to provide a new understanding of extended collision warning systems from an intelligent product-service systems perspective. The paper is well written and scientifically sound. I recommend the paper for publication in its present form.
Reviewer 4 Report
The manuscript presents a conceptual model of the extended collision warning system. It was clearly presented and well organized. In Sections 2 and 3, there are only definitions or concepts. Are there any equations for different models? And what are the advantages and disadvantages of different models in the literature? In Section 4, validation or comparison with other researchers' studies is suggested. What is the difference between the case study results in this manuscript with previous research? What is the reason for the difference?
